# Novel scFv against Notch Ligand JAG1 Suitable for Development of Cell Therapies toward JAG1-Positive Tumors

**DOI:** 10.3390/biom13030459

**Published:** 2023-03-02

**Authors:** Gabriela Silva, Ana F. Rodrigues, Susana Ferreira, Carolina Matos, Rute P. Eleutério, Gonçalo Marques, Khrystyna Kucheryava, Ana R. Lemos, Pedro M. F. Sousa, Rute Castro, Ana Barbas, Daniel Simão, Paula M. Alves

**Affiliations:** 1iBET, Instituto de Biologia Experimental e Tecnológica, Apartado 12, 2781-901 Oeiras, Portugal; 2Instituto de Tecnologia Química e Biológica António Xavier, Universidade Nova de Lisboa, Av. da República, 2780-157 Oeiras, Portugal

**Keywords:** JAG1, Notch signaling, single-chain variable fragment, chimeric antigen receptor, cytotoxic activity, cell therapy

## Abstract

The Notch signaling ligand JAG1 is overexpressed in various aggressive tumors and is associated with poor clinical prognosis. Hence, therapies targeting oncogenic JAG1 hold great potential for the treatment of certain tumors. Here, we report the identification of specific anti-JAG1 single-chain variable fragments (scFvs), one of them endowing chimeric antigen receptor (CAR) T cells with cytotoxicity against JAG1-positive cells. Anti-JAG1 scFvs were identified from human phage display libraries, reformatted into full-length monoclonal antibodies (Abs), and produced in mammalian cells. The characterization of these Abs identified two specific anti-JAG1 Abs (J1.B5 and J1.F1) with nanomolar affinities. Cloning the respective scFv sequences in our second- and third-generation CAR backbones resulted in six anti-JAG1 CAR constructs, which were screened for JAG1-mediated T-cell activation in Jurkat T cells in coculture assays with JAG1-positive cell lines. Studies in primary T cells demonstrated that one CAR harboring the J1.B5 scFv significantly induced effective T-cell activation in the presence of JAG1-positive, but not in JAG1-knockout, cancer cells, and enabled specific killing of JAG1-positive cells. Thus, this new anti-JAG1 scFv represents a promising candidate for the development of cell therapies against JAG1-positive tumors.

## 1. Introduction

Notch signaling is a cell-to-cell communication pathway composed of four Notch receptors (NOTCH1–4) and five ligands (DLL1, DLL3, DLL4, JAG1, and JAG2) that is crucial for embryonic development and postnatal tissue homeostasis [1]. Its dysregulation, particularly due to the overexpression of JAG1, is detected in diverse cancers and implicated in various hallmarks of cancer [2,3,4,5]. Increased expression of JAG1 is found in various types of cancer (e.g., brain, breast, colorectal, endometrial, gastric, hepatocellular, and prostate) and correlates with poor prognosis [6,7,8]. JAG1 overexpression in tumors contributes to tumor growth, metastasis, recurrence, and drug resistance by promoting cancer cell survival, proliferation, metastasis, cancer stem cell expansion, and tumor-associated angiogenesis, and by inhibiting tumor-specific immunity [7,8]. Importantly, JAG1 is a target of several oncogenic pathways (e.g., Wnt, TGFβ, and NF-kβ) [2,5,7,9]. The multifunctional role of JAG1 in cancer biology supports the development of anti-JAG1 therapies for the treatment of aggressive tumors. JAG1-targeting therapeutics are expected to provide clinical benefit by impairing therapy-enriched cancer stem cells and reducing metastasis and relapse. Anti-JAG1 antibodies (Abs) have been developed [9,10,11,12] and shown to have antitumor activity in in vivo models of triple-negative breast, ovarian, and colorectal cancers with no evident toxicity [9,11,12]. The antitumorigenic effects and safety profiles of these Abs show that JAG1 targeting is a promising therapeutic strategy against aggressive tumors expressing JAG1. The development of immunotherapies based on cells expressing chimeric antigen receptors (CARs) against JAG1 would increase the therapeutic potential of targeting JAG1 in cancer treatment.

Engineering immune cells to express CARs (synthetic receptors that redirect immune cells to recognize and eliminate cells expressing a target antigen) has been consolidated as a breakthrough treatment for hematological cancers [13] and is under investigation to target solid tumors, infectious diseases, and autoimmune disorders [14,15,16,17,18]. T cells have been the main type of immune cells used to generate CAR (CAR-T) cells, being particularly attractive due to their ability to expand and persist in vivo, sustaining medium- to long-term therapeutic effects [16,19,20]. CAR-T cells are the most advanced cell transfer immunotherapy adopted in clinics and comprise about half of the pipeline of cancer cell therapy worldwide [21].

Structurally, CARs comprise an extracellular domain that recognizes and binds to the target antigen, most commonly a single-chain variable fragment (scFv) of a monoclonal Ab, and an intracellular domain that signals for cell activation, containing one or two co-stimulatory subdomains (second- or third-generation design) plus the activation subdomain of the T-cell receptor complex, CD3ζ [22]. Both domains are linked via a hinge and a transmembrane region. Additional designs have been developed providing features such as interleukins secretion and other functionalities to improve CAR-T cell proliferation, activation, penetration of tumor microenvironment, persistence, and tumor specific-recognition and killing activity [16,18,22,23].

Due to the relevant role of JAG1 expression in various aggressive cancers, in this study, we aimed to identify novel specific anti-JAG1 scFvs with the potential for the development of therapeutic products, being either in the antibody format or in a cell modality setting. Here, we report the identification of two unique scFv Ab fragments from phage display libraries that specifically bind to cellular JAG1 in the IgG1 format and the generation and characterization of anti-JAG1 CAR constructs containing these scFvs. Studies using T cells that expressed these CARs demonstrated that one anti-JAG1 CAR induced effective T-cell activation in the presence of JAG1-positive but not JAG1 knockout tumor cells as well as the specific killing of JAG1-positive tumor cells. These findings indicate that one of these novel scFvs might be a good candidate for the development of cell therapies targeting JAG1-positive tumor cells. In light of this, these new anti-scFvs and generated anti-JAG1 CARs were made available in the Addgene repository for further exploitation and development and will hopefully enrich the toolbox of cancer therapies targeting JAG1.

## 2. Materials and Methods

### 2.1. Cell Lines and Culture Conditions

CHO-K1 (ATCC, Manassas, VA, USA, CCL-61) and HEK293E6 [24] cells were cultured as described in [25], and HEK293T (ATCC, CRL-3216) cells were cultured as described in [26]. HepG2 (ATCC, HB-8065) and Jurkat (ATCC, TIB-152) cells were cultured in RPMI 1640 (#61870) supplemented with 10% (*v*/*v*) heat-inactivated fetal bovine serum (FBS, #10270-106) and 100 μg/mL penicillin and streptomycin (#15140-122) (all from Gibco, Paisley, UK) at 37 °C in a humidified atmosphere with 5% CO_2_. All cells were tested for the absence of mycoplasma by Eurofins Genomics Europe Sequencing (Constance, Germany).

### 2.2. Generation, Production, Purification, and Characterization of Anti-JAG1 Monoclonal Abs

scFvs were selected by phage display using the human Tomlinson I+J libraries (Cambridge, UK, #ReIn_0017) the as target antigen; the recombinant human (rh) JAG1-EGF3-Fc (comprising the MNNL and DSL to EGF3 regions responsible for ligand binding to Notch receptors [27]) was selected as target antigen as described in [28]. The variable light (VL) and variable heavy (VH) regions of the unique scFvs that bound rhJAG1-Fc proteins but not the Fc control protein (all generated in house [28]) were synthesized and inserted into pCDNA3 vectors containing human Ig kappa or IgG1 constant regions for the light chain (LC) and heavy chain (HC), respectively (GenScript, Rijswijk, Netherlands). Anti-JAG1 Abs were produced in HEK293E6 cells as previously described [28]. The secreted Abs were purified by affinity chromatography (Mab Select SuRe, Cytiva, Marlborough, MA, USA) followed by size exclusion chromatography (Superdex 200 Prep Grade, Cytiva). Purified Abs were analyzed by sodium dodecyl sulphate-polyacrylamide gel electrophoresis (SDS-PAGE) and size exclusion-high-performance liquid chromatography (SE-HPLC), using an XBridge BEH 450Å column (Waters, Milford, MA, USA) as described in [29]. Protein concentration was determined by absorbance at 280 nm combined with the specific Ab extinction coefficients. Purity was estimated from SDS-PAGE and SE-HPLC [29]. Anti-JAG1 Abs reactivity, specificity, affinity, and blocking activity were evaluated with enzyme linked immunosorbent assay (ELISA), surface plasmon resonance (SPR), flow cytometry, and Notch reporter assays, as previously described [28]. For details on SPR, see the Appendix A.

### 2.3. Generation of CHO-k1 Stable Cells Overexpressing hJAG1 and HepG2 Cells Knocked Out for Endogenous JAG1

CHO-k1 cells overexpressing hJAG1 at the cell surface (CHO-k1/rhJAG1) were established by transfection of CHO-k1 cells with pFUSE vector (InvivoGen, Toulouse, France, pfuse-hg1fc1) containing the full-length hJAG1 cDNA (Sino Biological, Beijing, China, HG11648), essentially as described in [28]. The method of the generation of control CHO-k1 cells was previously reported [25].

For the generation of human HepG2 cells lacking JAG1 expression (HepG2/JAG1^ko^) and the respective control cells expressing JAG1 (HepG2/Ctr^ko^), HepG2 cells were transfected with JAG1 CRISPRCas9 KO vector-sc-400208-KO-2 or control CRISPRCas9 KO vector-sc-418922 (Santa Cruz Biotechnology, Dallas, TX, USA), separately, both expressing green fluorescent protein (GFP) using GeneJuice (Merck-Millipore, Kenilworth, NJ, USA, #70967) according to the manufacturer’s instructions. After 2 days, the GFP^+^ cell populations from both transfections were flow-cytometry-sorted on a BD FACSAria IIu (BD Biosciences, San Jose, CA, USA) and further cultured until they reached 9 million cells. Then, cells transfected with JAG1 CRISPRCas9 KO vector were stained with commercial antihuman JAG1 Ab, and the JAG1-negative population was sorted to obtain HepG2/JAG1^ko^ cells.

Flow cytometry sorting experiments were conducted at the Flow Cytometry & Antibodies Unit at Instituto Gulbenkian de Ciência, Lisbon, Portugal. JAG1 expression on the generated cells was confirmed by Western blotting.

### 2.4. CAR Lentiviral Constructs and Production and Titration of CAR Lentiviral Particles

CAR molecules were cloned into a lentiviral transgene derived from pRRLSIN.cPPT.PGK-GFP.WPRE: a self-inactivating lentiviral vector [30] encoding a green fluorescence protein (GFP) under the control of an internal human phosphoglycerate kinase 1 (hPGK) promoter and harboring a woodchuck hepatitis virus posttranscriptional regulatory element (WPRE) for mRNA stabilization. This plasmid was kindly provided by Didier Trono through the Addgene plasmid repository (Watertown, MA, USA, plasmid #12252). Before CAR cloning, this transgene was modified to substitute the human phosphoglycerate kinase 1 promoter by a human elongation factor-1 alpha (hEF1A) synthesized by Integrated DNA technologies (IDT, Newark, NJ, USA), and the GFP by mCherry reporter, amplified from pPUROmCherry [31], followed by an encephalomyocarditis virus internal ribosomal entry site (amplified from pIRESGALEO [32]), driving the expression of a GFP-zeocin fusion protein, made by PCR assembly using pRRLSIN.cPPT.PGK-GFP.WPRE as the template for GFP and pMONO-zeo-mcs (Invivogen, San Diego, CA, USA) for the Zeocin. This design was validated for bi-cistronic expression (mCherry and GFP) as well as positive selection in the presence of zeocin and then used as a cloning backbone for the different CAR constructs instead of mCherry.

A panel of CAR molecules was constructed in this study featuring a 2nd-generation design with the costimulatory domain of 4-1BB or a 3rd-generation design with both CD28 and 4-1BB costimulatory domains, as detailed in Appendix A. The following sequences were chosen for CAR construction: (i) human granulocyte-macrophage colony-stimulating factor (GMCSF) as signal peptide, (ii) human T-cell cluster of differentiation 8a (CD8A) for the hinge and transmembrane domain, (iii) human TNF receptor superfamily member 9 (TNFRSF9, herein, 4-1BB) as costimulatory domain in 2nd-generation CAR molecules, human cluster of differentiation 28 (CD28) as additional costimulatory domain in 3rd-generation CAR molecules, and (iv) scFv of anti-CD19 from FMC63 [33,34] with 218 linker [35] and the scFv of anti-JAG1 F1 or B5 Abs generated in this study with 218 linker (as in the case of CD19) or the (GGGGS)_3_ linker that attaches the VH and VL domains to one another in the scFv clones, which we selected from the phage display Tomlinson I+J libraries. These sequences were synthesized by IDT in suitable formats (plasmids or double-stranded DNA fragments) and used as templates for PCR assembly (plasmids) or directly for ligation (double-stranded DNA fragments) according to previously designed strategies to achieve the desired configurations, using an in-fusion cloning kit (Clontech Laboratories, Inc., Mountain View, CA, USA). Sequences encoding each of the scFvs were individually inserted into the respective CAR plasmids containing all the other CAR components during the last step of the cloning procedure. All final constructs were sequenced by Sanger sequencing. CARs plasmids maps and their respective sequences were deposited with the Addgene plasmid repository (plasmids: #194457, 194458 (anti-CD19 CARs), 194459, 194460, 194461, 194462, 194463, and 194464 (anti-JAG1 CARs)).

Lentiviral particles were produced by transient cotransfection of HEK293T cells using the 3rd-generation lentiviral packaging system [36]. Transducing units’ titer was determined by vector copy number quantitative PCR and complemented, for comparison purposes, with a titration protocol based on GFP expression, as described in [37]. Further details are given in the Appendix A. Primer and probe sequences used in the titration protocol are listed in Appendix A.

### 2.5. Generation of CAR-Jurkat T Cells

A suspension of Jurkat T cells was prepared in fresh RPMI medium with 10% FBS at a concentration of 2 × 10^6^ cells/mL containing 16 µg/mL of polybrene (Sigma Aldrich, St. Louis, MO, USA, H9268) and plated in 24-well plates at 500 µL/well. Serial dilutions of lentiviral supernatant were prepared in fresh medium and added to each well at 500 µL/well, resulting in a final concentration of polybrene of 8 µg/mL [38]. Cells were incubated at 37 °C; 24 h later, another 1000 µL of fresh medium was added to each well. At 48 h post-transduction, half of the cells were analyzed for transduction efficiency based on GFP and CAR expression at the cell surface as measured by flow cytometry. The remaining cells were recovered by centrifugation and resuspended in fresh medium containing 200 µg/mL of zeocin (Invivogen, San Diego, CA, USA, #ant-zn) at a concentration of 0.2 × 10^6^ cells/mL. Cells were kept under zeocin selection for 3 to 4 weeks with regular medium exchange and cell dilution, until reaching 1 × 10^6^ cells/mL. Thereafter, CAR-Jurkat T cells were reanalyzed for CAR expression by flow cytometry and Western blotting.

### 2.6. Generation of Primary CAR-T Cells

Peripheral blood mononuclear cells (PBMCs) were isolated from buffy coats from healthy human donor volunteers obtained from the Portuguese Blood Institute, as described in [39]. Pan T cells were purified from fresh recuperated PBMCs via magnetic negative selection using an EasySep Human T Cell Isolation Kit (StemCell Technologies, Vancouver, BC, Canada #17951) following the manufacturer’s instructions, typically yielding >95% pure—cell fractions as assessed by flow cytometry using anti-CD3, -CD4, and -CD8 Abs. Cells were immediately activated with T-cell TransAct anti-CD3/CD28 reagent matrix (Miltenyi Biotec, Bergisch Gladbach, Germany #130-128-758) according to the manufacturer’s instructions and cultured in TexMACS medium (Miltenyi Biotec, #130-097-196) with 50 IU/mL rhIL-2 (R&D Systems, Minneapolis, MN, USA, # 202-IL). After 2 days, cells were washed by centrifugation in TexMACS medium to remove anti-CD3/CD28 reagent, resuspended in RPMI medium with 10% FBS, and transduced via spinoculation in the presence of 8 µg/mL polybrene: for mock and each CAR, 0.5 × 10^6^ cells/200 µL of media in 12-well plates, we provided lentivirus at a multiplicity of infection of 2. The final volume was adjusted to 2 mL with RPMI medium with 10% FBS. T cells alone were also spinoculated to use as a control. After centrifugation, cells were incubated for 2 h before being diluted at 0.1 × 10^6^ cells/mL in expansion medium (RPMI with 10% FBS and 100 IU/mL rhIL2). The next day, the medium was replaced with fresh expansion medium and T cells were expanded at a density of 0.5–0.7 × 10^6^ cells/mL and cell numbers determined every 2 days. Following 5 days, CAR expression was confirmed by flow cytometry (GFP and scFv cell surface display) and Western blotting. T-cell phenotype and cell viability were also evaluated in cytometric assays. At 7 days post-transduction, the function of CAR-T cells was evaluated with activation, cytotoxicity, and cytokine release assays in RPMI medium with 10% FBS and without rhIL2.

### 2.7. Functional Characterization of CAR-T Cells

The CAR-induced activation of Jurkat T cells and primary T cells was evaluated with the flow cytometry detection of CD69 in GFP-positive cells, as recommended previously [40]. Nontransduced/parental and transduced Jurkat cells with mock or each CAR (effector cells) were cultured alone or with target cells expressing JAG1 (CHO-k1/rhJAG1 or HepG2/Ctr^ko^) and no target cells (CHO-k1/control or HepG2/JAG1^ko^). HepG2 cells (6 × 10^4^/cm^2^/600 µL/well in 24-well plates) and CHO-k1 cells (1 × 10^4^/120 µL/well in 96-well plates) were cultured for 2 days or 1 day, respectively. Then, the number of cells per well was determined. Effector cells were added to fresh medium at various effector-to-target (E:T) cell ratios. Primary T/CAR-T cells were cultured alone or cocultured with HepG2 target and nontarget cells, as described above, at a 3:1 E:T ratio. After 21 h, T cells were collected for CD69 expression analysis by flow cytometry. Experiments were performed in triplicate (96-well plates) or duplicate (24-well plates).

CAR-T-cell-mediated cytolytic activity was evaluated using an impedance-based real-time cell cytotoxicity assay (RTCA) xCELLigence system (Acea Biosciences, San Diego, CA, USA), as reported by others [41,42]. Briefly, HepG2/Ctr^ko^ (JAG1^+^) or HepG2/JAG1^ko^ cells (3 × 10^4^/150 µL/well) were cultured in 96-well E-plates (Acea Biosciences, #5232368001) and continuously monitored for nearly 46 h. Thereafter, 100 µL of medium was removed and replaced with 100 µL of fresh medium only or containing 1.5 × 10^5^ primary effector T cells (3:1 E:T ratio). Impedance measurements were then performed every 15 min for up to 76 h. All experiments were conducted in triplicate or quadruplicate. Percentage of cytolysis was calculated using the formula: [(impedance of HepG2 cells without effector cells − impedance of HepG2 cells with effector cells)/impedance of HepG2 cells without effector cells] × 100.

IL-2 and IFNγ levels were detected in cell culture supernatants from cocultures of primary T/CAR-T cells with HepG2/Ctr^ko^ (JAG1^+^) cells from the activation assays using Quantikine ELISA kits (R&D Systems, D2050 and DIF50C) following the manufacturer’s instructions.

### 2.8. Western Blotting and Flow Cytometry

Both assays were performed essentially as previously described [28,43]. For flow cytometry, except for protein L, the staining of cells was performed in ice-cold PBS containing 3% FBS. The surface expression of each CAR scFv was detected with protein-L [44] using ice-cold PBS with 3% bovine serum albumin as a buffer. After washing twice, cells were incubated in buffer with or without biotinylated-recombinant protein L (Pierce, #29997) (1 µg/mL/1 × 10^6^ cells, 45 min/4 °C). Then, cells were rinsed three times, incubated with Streptavidin-Alexa-647 (Molecular probes, S21374) (2 µg/mL/1 × 10^6^ cells, 30 min/4 °C), and washed. Flow cytometry was performed on a FACSCanto II system (BD Biosciences). Analysis was performed with FlowJo_v10.8.1 software (FlowJo, Ashland, OR, USA). For the full Ab list used in these assays, see Appendix A.

### 2.9. Statistical Analysis

The results are presented as the mean ± standard deviation (SD) or standard error of mean (SEM). Statistical significance was calculated with GraphPad Prism 9, version 9.3.1 (471), access on 2022) using ANOVA with Tukey’s multiple comparison test or the two-tailed Student’s *t*-test, as indicated in the corresponding figure legends. *p* < 0.05 was considered statistically significant.

## 3. Results and Discussion

### 3.1. Generation, Production, and Characterization of Anti-JAG1 Abs

To obtain Abs that specifically recognize human JAG1 with the potential to block JAG1–Notch signaling activation, we used rhJAG1-EGF3-Fc (encoding the MNNL to EGF3 domains) as the target antigen to select anti-JAG1 scFv fragments via phage display. After selection, we identified 55 clones that bound both rhJAG1-EGF3-Fc and rhJAG1-ECD-Fc (with the complete extracellular domain of hJAG1) proteins but not the Fc control protein [28] (Appendix A). Sequencing analysis identified 19 unique scFvs. Of these, seven clones without glycosylation sites in their complementarity-determining regions were reformatted into IgG1 Abs that were produced in HEK293E6 cells and purified in endotoxin-free conditions. The purified Abs, designated J1.B1, J1.B5, J1.D1, J1.F1, J1.F2, J1.F11, and J1.G4, presented the expected sizes of approximately 150 kDa for the full IgG, and 50 and 25 kDa for HC and LC, respectively (Appendix A), with purity > 95%.

To evaluate the binding ability of anti-JAG1 Abs for its cognate antigen, we performed a dose-dependent ELISA by incubating serial dilutions of anti-JAG1 Abs or isotype-matched control (Ctr) Ab [28] with fixed concentrations of rhJAG1-Fc proteins. As the negative control, Abs were also incubated with Fc protein. All seven anti-JAG1 Abs displayed binding to both JAG1 proteins but not to the Fc control (Figure 1a). J1.F1 showed the strongest binding, followed by J1.B5. The Abs J1.B1, J1.D1, and J1.F11 showed weak binding, while J1.F2 and J1.G4 showed very weak binding (Figure 1a). The calculated half-maximal effective concentration (EC_50_) of J1.B5 and J1.F1 to rhJAG1-ECD-Fc was found to be, on average, 77.8 and 13.5 nM, respectively (Table 1). The EC_50_ values of the other Abs were not possible to determine because Ab binding saturation was not achieved. No binding was observed with Ctr Ab. The specificity testing with ELISA under the same conditions showed that all anti-JAG1 Abs did not effectively bind to any other human Notch ligands (rhDLL1, rhDLL3, rhDLL4, and rhJAG2) (Figure 1b). Moreover, all the Abs except for J1.F1 were found to be cross-reactive to murine JAG1 (rmJAG1 presented more than 96% sequence identity to hJAG1) (Figure 1b). The measurement of the binding kinetics and affinity of J1.B5 and J1.F1 (the top binders) to rhJAG1-ECD-Fc by SPR (Figure 1c) revealed the dissociation rate constants for J1.B5 and J1.F1 as 52.2 and 18.6 nM, respectively (Table 1).

We next evaluated the ability of the anti-JAG1 Abs to bind hJAG1 expressed on the cell surface. The results of assays using CHO-k1 stable cells overexpressing hJAG1 (CHO-k1/rhJAG1) and control cells without hJAG1 (CHO-k1/Control) (Figure 2a and Appendix A) revealed that J1.B1, J1.B5, J1.D1, J1.F1, and J1.F11 Abs bound exclusively to CHO-k1/rhJAG1 cells (Figure 2a, left panels). Further assays using the cell lines representative of different cancers endogenously expressing various levels of JAG1 or not (Appendix A) showed that anti-JAG1 Abs bound solely to JAG1-expressing cells in accordance with JAG1 expression levels (Appendix A). J1.F1 displayed the strongest binding to endogenous JAG,1 followed by J1.B5 (Appendix A and Figure 2a, right panel). Specificity testing using HepG2 cells expressing JAG1 (HepG2/Ctr^ko^) and lacking JAG1 expression (HepG2/JAG1^ko^) (Figure 2b, upper panels and Appendix A) showed that J1.B1, J1.B5, J1.D1, J1.F1, and J1.F11 Abs bound to HepG2/Ctr^ko^ but not to HepG2/JAG1^ko^ cells (Figure 2b).

To evaluate the ability of Abs to block JAG1–Notch signaling activation, MCF-7 cells transfected with Notch-luciferase reporter were plated in wells precoated with rhJAG1-ECD-Fc to induce reporter activity. Cells were either not treated (NT) or treated with anti-JAG1 Abs, Ctr Ab, DAPT, or DMSO. Only J1.D1 caused a decrease of approximately 30% in Notch signaling induction by rhJAG1 compared with that of untreated control cells (*p* = 0.033) or cells treated with Ctr Ab (Figure 2c). DAPT abrogated JAG1–Notch signaling (*p* < 0.0001). These results suggest that J1.D1 binds to a region in JAG1 involved in binding to Notch receptors and is important for the activation of Notch signaling, in contrast with the other Abs, particularly J1.B5 and J1.F1 that bind best to cellular JAG1. The weak blocking activity of J1.D1 is likely due to its weak binding to hJAG1 (Figure 1a and Figure 2a,b). Of notice, there was no evidence of any toxicity associated with any of the Abs in any the assays that were carried out.

Collectively, these data show that we were able to select scFvs clones that specifically recognize cellular JAG1 in the monoclonal Ab format, two of them (the lead Abs J1.B5 and J1.F1) with two-digit nanomolar affinities and one with modest blocking activity (J1.D1).

### 3.2. Generation of Anti-JAG1 CARs and CAR-Jurkat T Cells

Following the findings described above, and considering the oncogenic role of JAG1 in various aggressive tumors [7,8] and the broad promise of immune cell therapies [15,21], we hypothesized that the scFvs of J1.B5 and J1.F1 (sequences provided in Appendix A) may be good candidates to engineer CAR immune cells with anticancer activity against JAG1-expressing tumor cells. Because CAR-T therapy is the most advanced therapy used in the clinics [21], we next generated anti-JAG1 CAR-T lentiviral constructs with B5 and F1 scFvs to explore this hypothesis. CAR-T cell function is highly dependent on CAR modular components and configuration [22]. Accordingly, six anti-JAG1 CAR constructs were generated featuring second- and third-generation (2G and 3G, respectively) designs in various formats and coexpressing a zeocin–GFP fusion protein for fluorescence detection and selection of CAR-T cells (Figure 3a): four anti-JAG1 CARs designated B5(VH-VL)-2G, B5(VH-VL)-3G, F1(VH-VL)-2G, and F1(VH-VL)-3G containing the B5 or F1 VH and VL regions and the (GGGGS)3 linker in the scFv configuration present in the clones selected by phage display that effectively bound the recombinant JAG1 proteins (Appendix A), with the assumption they would successfully recognize the respective target antigen in the context of a CAR protein; two CARs containing the VH and VL regions of the lead Ab binder J1.F1 and the 218 linker in the configuration of anti-CD19 CARs [45], designated F1(VL-VH)-2G and F1(VL-VH)-3G, to test whether they would lead to a superior CAR-T cell effector function.

Anti-CD19-2G and -3G CARs were also generated as negative controls (Figure 3a and Appendix A) as well as a mock (GFP) construct.

For a first characterization of the generated CAR constructs, we used Jurkat cells, a leukemic T cell line that does not express JAG1 (Appendix A), which is commonly used as a T-cell model for the primary screening and functional validation of new CARs [40]. Although Jurkat cells have no significant cytotoxic activity, they do secrete some cytokines such as primary T cells. Jurkat cells were transduced with each CAR or GFP (mock) lentiviral vectors. Nontransduced/parental cells were used as controls. All the generated CARs were expressed with the expected sizes and displayed their scFvs at the cell surface, as demonstrated by Western blotting and protein L binding by flow cytometry (Figure 3b,c). Interestingly, all 3G CARs yielded lower protein expressions. Thus, we successfully constructed six novel anti-JAG1 CARs.

### 3.3. Anti-JAG1 CAR-Jurkat T Cells Specifically Recognize JAG1-Expressing Cells

Upon binding target antigen, Jurkat cells upregulate the T-cell activation marker CD69 [40]. CHO-k1 cells expressing human proteins were used to screen CAR functionality in coculture assays with CAR-T cells [46]. These cells have high growth rates, form good monolayers, and do not detach in patches when confluent, allowing the easier recovery of suspension cells in coculture assays and the quick screening of antigen-specific responses in T cells. Accordingly, we first used CHO-k1/rhJAG1 (target) and CHO-k1/control (no target) cells to evaluate whether anti-JAG1 CARs specifically recognize the JAG1 cell antigen. Jurkat parental cells and cells stably expressing each CAR or mock were cultured alone (control condition) or with eitherJAG1-positive or -negative CHO-k1 cells at various E:T ratios. CD69 expression was evaluated via flow cytometry, as a marker of cell activation. As demonstrated in Figure 4a–c, all six anti-JAG1 CARs significantly increased CD69 expression in Jurkat cells cocultured with CHO-k1/rhJAG1 compared with that of cells cultured alone or with CHO-k1/control (*p* < 0.001), indicating these CARs specifically recognize rhJAG1. In addition, 3G CARs induced the highest antigen-specific activation of Jurkat cells. The anti-JAG1 CARs effects were specific because no significant changes in CD69 expression were observed in cells transduced with the mock (GFP) or negative control anti-CD19 CARs. Furthermore, contrary to 2G CARs, the anti-JAG1 3G CARs also significantly increased the CD69 expression in the Jurkat cells grown in the absence of JAG1 compared with that of parental control cells (*p* < 0.05). This indicates that these 3G CARs can elicit the autoactivation/tonic signaling of Jurkat cells.

We then performed similar assays using HepG2/Ctr^ko^ cells endogenously expressing JAG1 to evaluate whether anti-JAG1 CARs also recognize endogenous hJAG1. HepG2/JAG1^ko^ cells were used as the control (no target). Interestingly, the results from these assays revealed that only anti-JAG1 CARs comprising the B5 and F1 scFvs in the VH-VL configuration with the (GGGGS)_3_ linker led to the significant activation of Jurkat cells cocultured with HepG2/Ctr^ko^ cells compared with that of cultures with JAG1-negative cells (*p* < 0.02) (Figure 4d–f). Again, anti-JAG1 3G CARs induced the highest JAG1-mediated activation of Jurkat cells and tonic signaling (Figure 4d–f). The lack of activation for F1 2G and 3G CARs harboring the scFv in the VL-VH order with the 218 linker when cocultured with HepG2/Ctr^ko^, contrary to co-cultures with CHO-k1/rhJAG1, could have been due to the configuration of this scFv that induced binding affinity changes, making it bind solely to very high antigen-expressing cells. Conformational differences amongst native and recombinant hJAG1 proteins might also have accounted for these CARs’ activity differences, meaning that this scFv configuration does not effectively bind endogenous hJAG1.

As specified above, CAR-T cell function also depended on CAR elements and the order that dictates CAR protein expression, configuration, and ultimately the recognition of the target antigen, apart from the levels of the antigen expression in target cells [19,22,47]. Accordingly, the highest JAG1-mediated activation of Jurkat cells by the anti-JAG1 3G CARs (Figure 4), expressed at lower amounts compared with the respective 2G CARS (Figure 3b,c), was likely related with the different costimulatory domains present in 2G and 3G CARs (Figure 2a). The presence of two in-line costimulatory domains in 3G CARs may have rendered them more prone to JAG1-mediated T-cell activation. In addition, the presence of CD28, shown to induce different downstream pathways in CAR-T cells compared with 4-1BB [45], could have contributed to the highest antigen-mediated activation of Jurkat cells by the 3G CARs.

Importantly, no significant effects on cell viability were observed for any anti-JAG1 CARs, neither for Jurkat cells cultured alone nor with nontarget or target cells (Appendix A). Moreover, anti-CD19 CAR-Jurkat cells demonstrated increased CD69 expression after coculture with CD19-positive Raji cells (Appendix A), indicating specific activation and confirming that the anti-CD19 CAR constructs were functional.

Collectively, these results show that we were able to generate anti-JAG1 CARs capable of recognizing JAG1 and inducing JAG1-specific responses in Jurkat T cells.

### 3.4. scFv from Anti-JAG1 J1.B5 Ab, in a Third-Generation CAR Design, Effectively Activates Primary T Cells and Kills Cancer Cells in a JAG1-Dependent Manner

We next investigated the function of anti-JAG1 CARs that induced JAG1-mediated activation of Jurkat cells in primary T cells (Figure 5a). The T cells from three donors were transduced with each of these anti-JAG1 CARs, anti-CD19 CARs, or mock lentiviral vectors. Nontransduced T cells were also kept as controls. The transduction efficiencies of primary T cells with the various CARs varied from 7% to 28%, with the lowest and highest percentage of transduced T cells consistently achieved with anti-JAG1 B5(VH-VL)-2G and anti-JAG1 F1(VH-VL)-3G CARs, respectively (Figure 5b and Appendix A). All CAR-T primary cells displayed their scFvs at the cell surface, as demonstrated by protein L binding (Figure 5b). The immunophenotype of CAR-T and control T cells (parental and mock-transduced) showed all of them were CD45RA^+^CD45RO^+^CD62L^+^ (Figure 5c), suggesting a central memory T-cell phenotype [48]. However, CAR-T cells expressing the anti-JAG1 2G CARs presented lower levels (*p* < 0.05) of CD62L marker (associated with naïve/memory T-cell subsets) (Appendix A), suggesting they had a more differentiated central memory phenotype than the other CAR-T/T cells. In addition, a fraction of these CAR-T cells expressed higher levels (*p* < 0.05) of CD69 than all other T/CAR-T cells, indicating these 2G CARs induced tonic signaling of primary T cells (Appendix A). None of the generated CAR-T cells expressed PD1, a marker of exhausted T cells [49] (Appendix A). These results demonstrate that we generated primary CAR-T cells with a central memory phenotype, associated with superior persistence and antitumor immunity compared with those of effector memory T cells [48].

Examination of JAG1-mediated activation of primary CAR-T cells in coculture assays with HepG2 cells showed a consistent increase in the percent of CD69-positive T/CAR-T cells compared with that of the respective cells cultured alone (Figure 5d and Appendix A). Nevertheless, significant effective T-cell activation (*p* < 0.0001) was observed only in T cells expressing anti-JAG1 B5(VH-VL)-3G CAR cocultured with JAG1-expressing HepG2 cells (HepG2/Ctr^ko^) (Figure 5d–f), demonstrating the antigen-specific activity of this CAR. Contrary to data obtained in Jurkat cells, anti-JAG1 2G CARs induced CD69 expression in primary T cells grown in the absence of JAG1 (Figure 5d–f), revealing that these CARs caused tonic signaling in these cells. It was shown that CAR-mediated tonic signaling in Jurkat cells is predictive of tonic signaling in primary T cells [40]. The different tonic signaling we obtained in this study with these novel anti-JAG1 CARs amongst Jurkat and primary T cells might be related to the biological features of the Jurkat T cells used herein. Nevertheless, this suggests that data obtained on Jurkat cells should always be thoroughly evaluated in follow-up studies on primary T cells.

Our investigation of the antigen-specific killing activities of anti-JAG1 primary CAR-T cells via a real-time cytotoxicity assay (RTCA) revealed that anti-JAG1 B5(VH-VL)-3G CAR-T cells from different donors effectively killed JAG1-expressing HepG2 cells (*p* < 0.0001) and had no killing activity against JAG1-negative HepG2 cells (Figure 5g and Appendix A). In contrast, primary T cells from two donors expressing anti-JAG1 2G CARs exerted some nonspecific cell-killing effects (Figure 5g and Appendix A). No significant killing of either HepG2 cells was observed with nontransduced primary T cells or other CAR-T cells.

The measurement of cytokine secretion by T/CAR-T cells cocultured with JAG1-expressing HepG2 cells showed significant increases in IL-2 (*p* = 0.0210) and IFN-γ (*p* = 0.0136) only for anti-JAG1 B5(VH-VL)-3G CAR-T cells compared with those of control cells (Figure 5h). Furthermore, anti-JAG1 2G CAR-T cells, which suffer from tonic signaling (Figure 5d–f), also presented a trend toward higher IFNγ secretion (Figure 5h).

Overall, these data revealed that primary T cells expressing the anti-JAG1 B5(VH-VL)-3G CAR specifically recognize JAG1, become activated, and effectively and specifically kill JAG1-expressing cancer cells in vitro. Thus, this novel anti-JAG1 CAR is expected to have antitumor activity against JAG1-positive cancers.

Like most other tumor-promoting proteins, JAG1 is also found in some normal tissues [7,10]. Because of this, the in vivo administration of anti-JAG1 B5(VH-VL)-3G CAR-T cells can also attack noncancer cells and cause on-target, off-tumor toxicity. Thus, the in vivo antitumor potential of anti-JAG1 B5(VH-VL)-3G CAR must be evaluated using engineered T cells with combinatorial antigen-recognition circuits allowing their activation exclusively upon the recognition of JAG1-expressing cancer cells. Synthetic Notch (synNotch) receptors, which function by inducing the expression of effector proteins upon antigen recognition, have been introduced into CAR-T cells and have been shown to augment CAR-T-cells-specific tumor recognition and promote persistent antitumor activity against different cancers [20,23,50,51]. Accordingly, engineered T cells equipped with synNotch receptors binding other antigens present in JAG1-expressing cancers that drive the inducible expression of anti-JAG1 B5(VH-VL)-3G CAR may present a good approach to evaluate the antitumor efficacy of this CAR and the therapeutic potential of this novel specific anti-JAG1 scFv in the cell therapy setting.

CAR-T cell therapies have shown remarkable clinical success against hematologic tumors such as leukemia, lymphoma, and multiple myeloma, but limited efficacy in solid tumors, even though early phase clinical trials in various solid tumors have shown encouraging responses rates and favorable safety profiles [17,52]. The efficacy of CAR-T cells against solids tumors is hampered by target antigen heterogeneity as well as the immunosuppressive tumor microenvironments, amongst other restricting obstacles [17,52]. Still, because of the unmet clinical need to find more efficacious therapies against aggressive solid tumors, numerous efforts are ongoing to develop novel strategies that may eventually defeat the existing biologic barriers to CAR-T cell progress in such tumors [16,18]. These include strategies to induce the expression of CAR target antigens in tumor cells, the design of new CAR-T agents targeting the tumor microenvironment (e.g., CAR-T cells targeting tumor stroma, extracellular matrix, suppressive cells, and cytokines/chemokines), and CAR-T-boosting vaccines [18,53]. Because these strategies have demonstrated that it may be possible to overcome the mentioned biological barriers and increase the efficacy of CAR-T therapy for the treatment of solid tumors, our anti-JAG1 scFvs can be applied to engineered T cells using some of these approachs and combinatorial antigen recognition circuits (e.g., SynNotch system) or dual-target CAR approaches [23] to improve both their efficacy and safety by minimizing on-target, off-tumor effects. With this in mind and considering our results, we plan to engineer CAR-T cells with anti-JAG1 B5(VH-VL)-3G CAR equipped with some of the newly developed strategies to evaluate this CAR´s antitumor potential in relevant mouse tumor models and further explore the potential of the scFv identified herein for the development of cell therapies against solid tumors.

## 4. Conclusions

Here, we selected anti-JAG1 scFv binders from phage display libraries. The characterization of the resulting Abs with ELISA, SPR, and flow cytometry allowed the identification of specific anti-JAG1 Abs, two of which showed two-digit nanomolar affinities without blocking activity (J1.B5 and J1.F1) and one showed low binding and modest blocking activity (J1.D1). Given their specificities and nanomolar affinities and considering the oncogenic role of JAG1 as well the great potential of cell therapies against cancer, six anti-JAG1 CARs containing the scFvs from J1.B5 or J1.F1 Abs were generated featuring second- or third-generation designs. The characterization of these CARs in Jurkat T cells showed that all CARs were expressed and four of them were able to induce JAG1-mediated Jurkat-cell activation. The further characterization of these CARs revealed that engineered primary CAR-T cells expressing the J1.B5 Ab scFv in a third-generation design became activated only in the presence of JAG1-expressing cells and effectively and specifically killed JAG1-positive cancer cells. Based on these data, this novel anti-JAG1 scFv presents as a promising candidate for the development of cell therapies against JAG1-positive tumors. It is plausible that it can expand the toolbox of scFvs with the potential for the development of therapies against aggressive solid tumors.

## Figures and Tables

**Figure 1 biomolecules-13-00459-f001:**
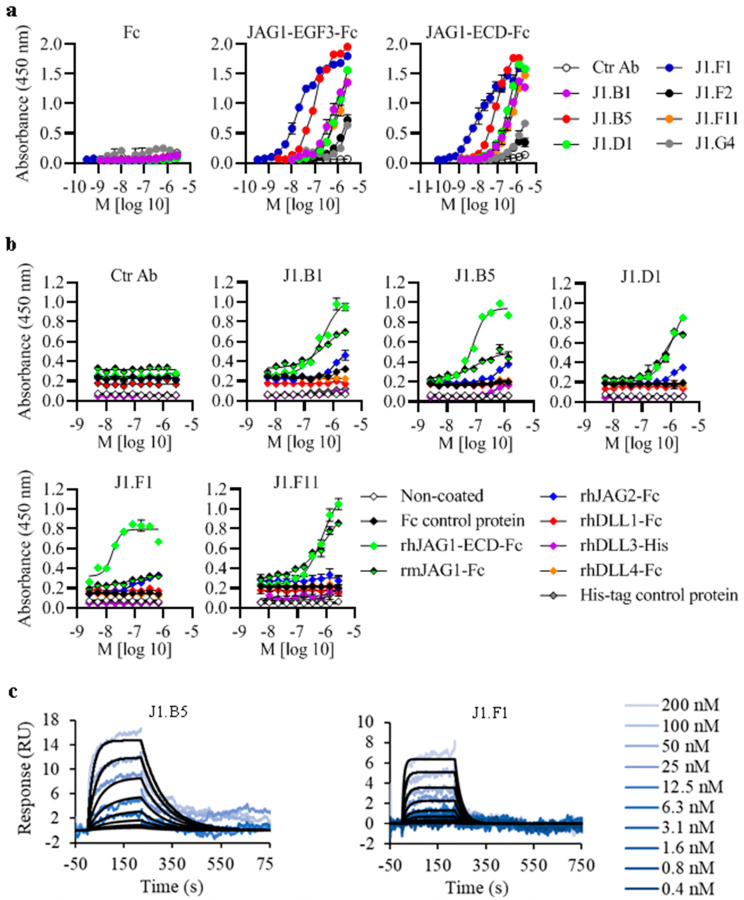
Anti-JAG1 Abs binding ability, specificity, and affinity for recombinant human (rh) JAG1 proteins. (**a**) Ability of anti-JAG1 Abs to recognize rhJAG1 proteins. rhJAG1-EGF3-Fc (with ECD up to the EGF3 domain), rhJAG1-ECD-Fc (with complete ECD), and Fc (negative control) protein were immobilized in 96-well plates (5 µg/mL in PBS, 16 h, and 4 °C) and incubated with serial dilutions of anti-JAG1 Abs or isotype-matched negative control Ab (Ctr Ab) (starting from 200 µg/mL). Binding was evaluated with ELISA. (**b**) Binding specificity of anti-JAG1 Abs. Abs binding to rhJAG1-Fc proteins was tested for binding to other human Notch ligands (rhDLL1, rhDLL3, rhDLL4, and rhJAG2) and murine JAG1 (rmJAG1) as in (**a**). Noncoated wells and wells coated with Fc or with an irrelevant His_6_-tagged protein were used as negative controls. Graphs in (**a**,**b**) are representative of three independent assays each performed in duplicate. (**c**) SPR sensorgrams of rhJAG1–ECD–Fc interactions with the lead anti-JAG1 Abs J1.B5 and J1.F1. Representative curves from one assay run with three technical replicates are shown.

**Figure 2 biomolecules-13-00459-f002:**
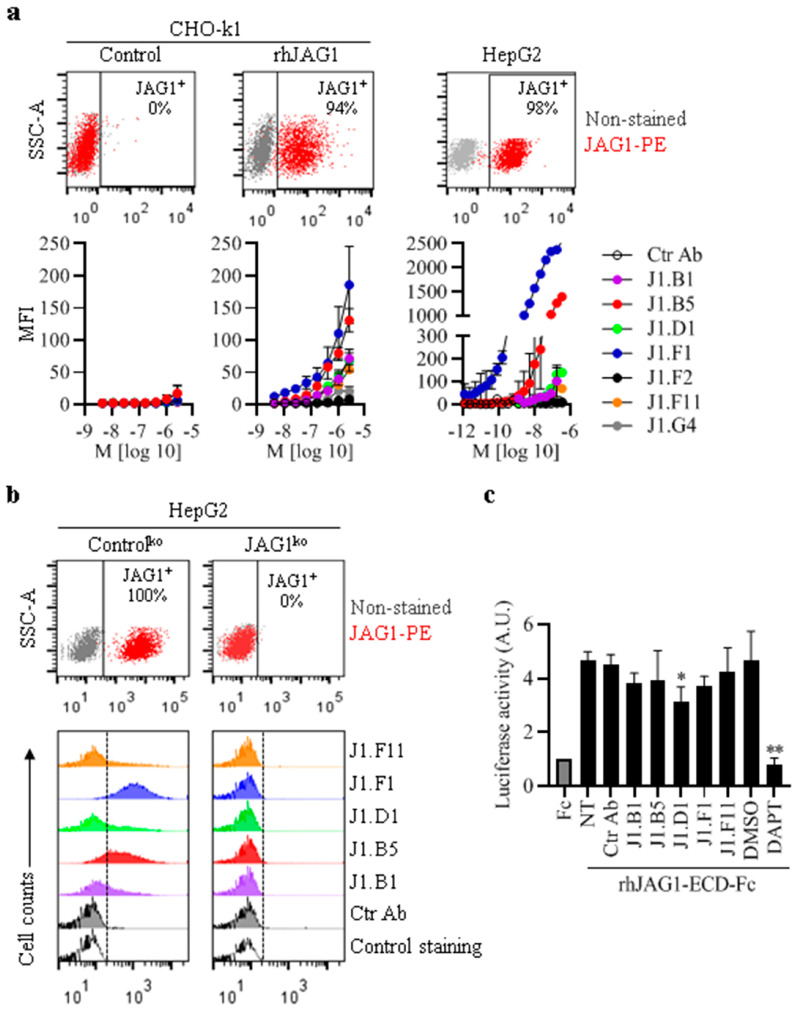
Anti-JAG1 Abs binding ability and specific to cellular human JAG1 and blocking activity. (**a**,**b**) Flow cytometry analysis showing anti-JAG1 Abs bind exclusively to JAG1-positive cells. (**a**) Dose–response binding of anti-JAG1 Abs or Ctr Ab to CHO-k1 cells overexpressing hJAG1 on the cellular surface (CHO-k1/rhJAG1), CHO-k1 control cells (CHO-k1/control), and HepG2 cells with endogenous JAG1. Graphs show mean fluorescence intensity (MFI) (±SEM) of two independent assays. Upper cytometric plots show JAG1 expression. (**b**) Binding of 2700 nM of anti-JAG1 cell binders identified in (**a**) and Ctr Ab to HepG2 JAG1-positive (HepG2/control^ko^) and HepG2 JAG1-negative (HepG2/JAG1^ko^) cells. Control staining indicates cells stained with secondary antihuman IgG (H+L)-A488 alone. (**c**) Effect of anti-JAG1 Abs in JAG1-mediated Notch reporter activation. Cells transfected with the Notch firefly luciferase reporter along with a vector encoding renilla luciferase were cultured in wells precoated with Fc control protein or rhJAG1-ECD-Fc in the absence (NT) or presence of either anti-JAG1 Abs, Ctr Ab (20 µg/mL each), the pan-Notch inhibitor DAPT (5 μM), or dimethyl sulfoxide (DMSO, DAPT vehicle). After 36 h, luciferase activities were determined with dual luciferase assay. The graph shows luciferase activity compared with that of cells cultured in the presence of Fc (mean ± SD) from five independent assay, with three or four replicates for each condition. *, *p* = 0.05 vs. control NT cells; **, *p* < 0.05 vs. cells treated with DMSO, as determined by two-tailed paired *t*-test.

**Figure 3 biomolecules-13-00459-f003:**
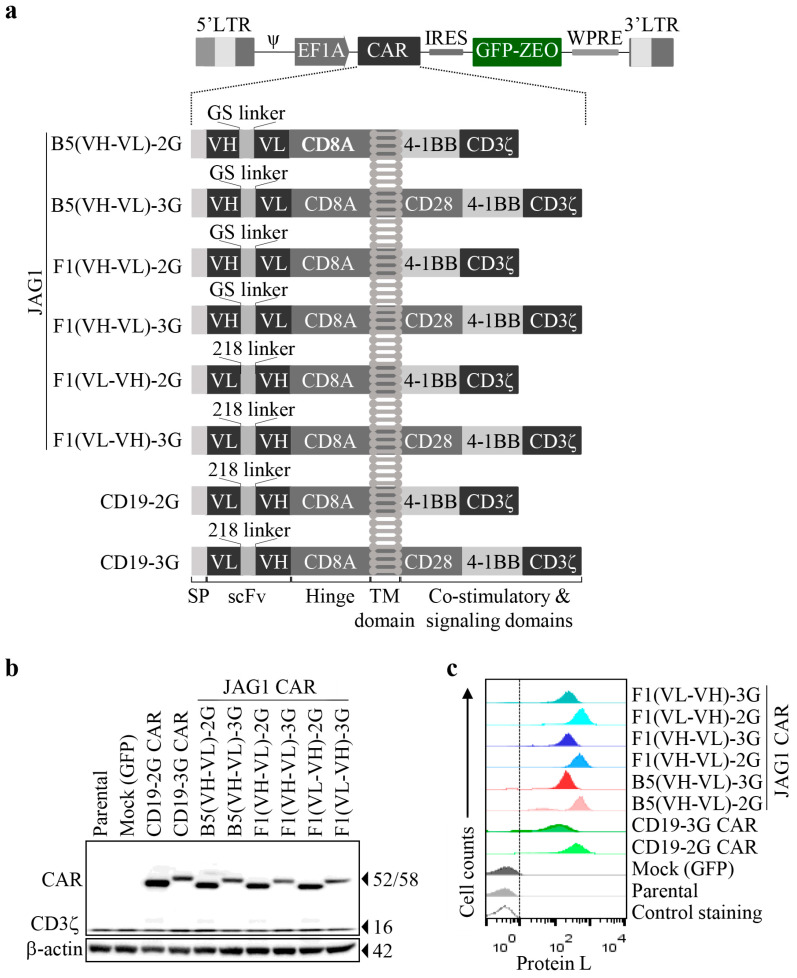
Design of CAR constructs and evidence of CAR expression in transduced Jurkat T cells. (**a**) Schematic representation of functional regions of generated lentiviral CAR constructs used to obtain engineered T cells. After lentivirus transduction, effector cells expressed one of the depicted CARs consisting of the GMCSF signal peptide (SP) and the variable light (VL) and variable heavy (VH) chains of the scFv from J1.B5 or J1.F1 Abs (anti-JAG1 B5 or F1 CARs) or from FMC63 (anti-CD19 CARs), the flexible linkers 218 or (GGGGS)_3_, the hinge and transmembrane (TM) domain of CD8A, the CD28 and/or 4-1BB as costimulatory domains, and the CD3ζ cytoplasmic domain of T cells; zeocin–GFP fusion protein enabled detection and selection of CAR-transduced cells. Ψ (psi), sequence for packaging the viral genome into the capsid. (**b**,**c**). CARs expression on Jurkat cells. Cells were transduced with control (mock) or one of the different CAR lentiviral vectors. (**b**) Immunoblotting analysis of CAR expression 72 h post-transduction. Detection of β-actin was used as loading control. (**c**) Flow cytometry plots showing surface expression of CARs in Jurkat cells after zeocin selection of transduced cells detected by binding to protein-L. Nontransduced/parental and mock-transduced cells were used as controls.

**Figure 4 biomolecules-13-00459-f004:**
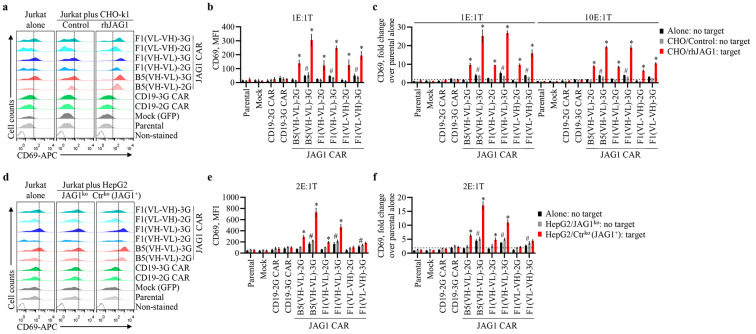
Antigen-specific activation of anti-JAG1 CAR-Jurkat T cells. (**a**–**f**) Parental Jurkat cells, mock-transduced cells, and zeocin-selected transduced cells anti-JAG1 or control anti-CD19 CARs were cultured alone or with JAG1-negative cells CHO-k1/control or HepG2/JAG1^ko^ (no-target cells) or the JAG1-positive cells CHO-k1/rhJAG1 or HepG2/Ctr^ko^ (target cells) at the indicated effector to target ratio (E:T). After 21 h, expression of the T-cell activation marker CD69 in Jurkat cells was evaluated via flow cytometry. Mock and CAR-Jurkat cells were gated on GFP signals. (**a**,**d**) Representative flow cytometry plots of CD69 expression in Jurkat cells’ monocultures or cocultures with the indicated cells using an E:T ratio of 1:1 and 2:1, respectively. Horizontal dashed lines mark CD69 induction threshold relative to control parental cells. (**b**,**e**) Bar graphs show quantification of CD69 expression (mean ± SEM) of three and four independent experiments performed with CHO-k1 and HepG2 cells, respectively. MIF, mean fluorescence intensity. (**c**,**f**) Mean fold values (±SEM) of CD69 expression relative to parental Jurkat cells cultured alone at the tested E:T ratios from at least three independent assays. The minimal threshold fold increase (horizontal doted lines in (**c**,**f**)) was set to 2. *, significant difference (*p* < 0.05) between the anti-JAG1 CAR-Jurkat cells cocultured with target cells expressing JAG1 and control parental Jurkat cells cultured alone. #, significant difference (*p* < 0.05) between the indicated groups cultured alone or cocultured with JAG1-negative cells and parental control cells cultured alone. Data were analyzed using 2-way ANOVA with Tukey´s multiple comparisons test.

**Figure 5 biomolecules-13-00459-f005:**
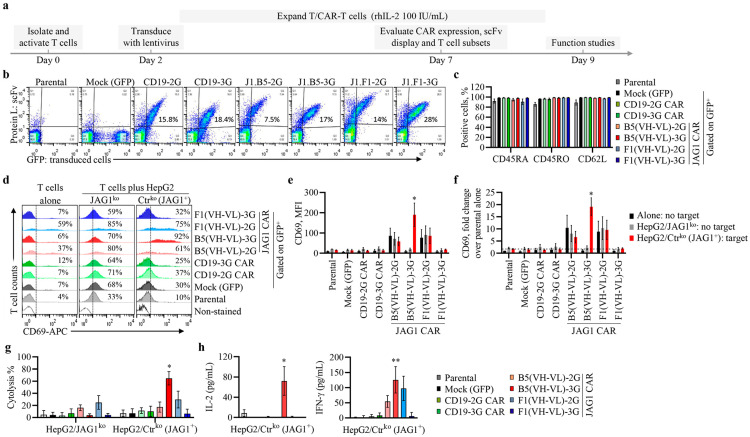
Expression, activation, and killing activities of anti-JAG1 CARs in primary T cells. (**a**) Schematic diagram of CAR-T cells’ generation and characterization from three different donors. (**b**) Surface expression of CARs in transduced CAR-T cells, defined as GFP^+^ cells, by protein L binding detection. Nontransduced and mock-transduced cells were used as negative controls. Data are representative from assays with cells from three donors (**c**) Mean percentage (±SEM) of the indicated T-cell phenotype markers in nontransduced parental cells, gated GFP-positive mock, or CAR-T cells from three donors. (**d**–**f**) Expression of T-cell activation marker CD69 in T/CAR-T cells from three donors cultured alone or with HepG2 JAG1-negative (HepG2/JAG1^ko^) or JAG1-positive (HepG2/Ctr^ko^ (JAG1^+^)) cells at a 3:1 E:T ratio. (**d**) Representative flow cytometry plots of CD69 expression obtained in one of these assays after 21 h of culture. Graphs in (**e**,**f**) show mean fluorescence intensity (MFI) and fold-change values of CD69 (mean ± SEM) from these assays, respectively. *, significant differences (*p* < 0.05) between the indicated CAR-T cells cocultured with JAG1-positive and JAG1-negative cells or with parental cells cultured alone. (**g**) CAR-T cell cytotoxicity to JAG1-expressing HepG2 cells. HepG2/Ctr^ko^ and HepG2/JAG1^ko^ were seeded in 96-well E-plates and cultured with or without T/CAR-T cells and growth monitored over time with RTCA. The graph shows mean percentage (±SEM) of cytolysis of HepG2 cells by T/CAR-T cells from three donors relative to HepG2 cells cultured alone after 76 h of culture. *, *p* < 0.05 vs. HepG2/Ctr^ko^ cells cultured with parental T cells. (**h**) Cytokine secretion (mean ± SEM) in cell culture supernatants from cocultures of JAG1-positive cells with the indicated T/CAR-T cells from three donors after 21 h of coculture. *, *p* = 0.0210 and **, *p* = 0.0136 vs. co-cultures with parental T cells. Data were analyzed using 2-way ANOVA (**e**–**g**) and 1-way ANOVA (**h**) with Tukey´s multiple comparisons test.

**Table 1 biomolecules-13-00459-t001:** Binding properties of anti-JAG1 B5 and F1 Abs to JAG1. EC_50_ values for rhJAG1 and endogenous cellular JAG1 from HepG2 cells were determined by ELISA and flow cytometry, respectively. Affinity constants were estimated with SPR assays with JAG1-ECD-Fc protein. Data are mean values ± SD (*n* = 3).

Ab	ELISA	SPR			Flow Cytometry
	EC_50_ (nM)	K_D_ (nM)	*k_a_* (s^−1^)	*k_d_* (M^−1^ s^−1^)	EC_50_ (nM)
J1.B5	77.8 ± 4.06	52.2 ± 6.65	2.03 × 10^5^ ± 2.37 × 10^4^	1.06 × 10^−2^ ± 1.31 × 10^−3^	336.0 ± 147
J1.F1	13.5 ± 2.74	18.6 ± 8.14	1.00 × 10^6^ ± 1.60 × 10^5^	1.86 × 10^−2^ ± 4.18 × 10^−3^	5.74 ± 0.868

## Data Availability

Data are presented within the article and Appendix A. Maps and sequences of CAR plasmids were deposited with the Addgene plasmid repository with numbers 194457, 194458, 194459, 194460, 194461, 194462, 194463, and 194464.

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
