# Peer review of "Novel scFv against Notch Ligand JAG1 Suitable for Development of Cell Therapies toward JAG1-Positive Tumors"

_biomolecules, 2023, doi:10.3390/biom13030459_

Round 1
Reviewer 1 Report
The topic presented by the authors is of interest.
FDA-approved CAR T-cell therapies relate mostly to B-cell lymphomas, also referred to as liquid tumors.
Solutions for applications of CAR T-cell therapies to "solid tumors" await more technological advancement. Although the authors mention "solid tumors" in the introduction, they are casual about developing solutions through their technology. The problem with solid tumors is that the distribution of potential targets are distributed unevenly on the surface of the tumor and other more interior regions of the tumor.
The solution presented here may target JAG1-expressing tumors but may lack success on 3D tumors.
The work is impressive in the sense of strategy. The result is expected, meaning is not really a novel avenue to solve the problem. I also imagine the authors could have developed ScFv proteins for any other type of target.
I found the introduction lacks citations. The sentences between lines 42 to 49. Similarly between 65 to 70.
The authors refer much to citation #26, published by them. Righteously so.
I would like to see a few more references to previous articles through the methods as well. But I understand that is hard to do sometimes.
In the conclusions authors write:
4. Conclusions 558
Using a recombinant truncated human JAG1 protein, containing the JAG1 regions 559
involved in Notch binding/activation, we selected anti-JAG1 scFv binders from phage dis- 560
play libraries.
This sentence seems to come from things the authors did in reference#26.
If this is correct they did not do this work on this manuscript.
Author Response
Comment 1: The solution presented here may target JAG1-expressing tumors but may lack success on 3D tumors.
Reply to comment 1: As correctly stated by the reviewer CAR-T cell therapies have shown remarkable clinical success against hematologic tumors such as leukemia, lymphoma, and multiple myeloma but limited efficacy in solid tumors, even though early phase clinical trials in various solid tumors showed encouraging responses rates and favorable safety profiles [1,2]. As pointed by the reviewer the efficacy of CAR-T cells against solids tumors is hampered by target antigen heterogeneity as well as the immunosuppressive tumor microenvironments, amongst other restricting obstacles. Still, because of the unmet clinical need to find more efficacious therapies against aggressive solid tumors, numerous efforts are ongoing to develop novel strategies that may eventually defeat the existing biologic barriers to CAR-T cell progress in such tumors [3,4]. These include strategies to induce the expression of CAR target antigens in tumor cells, the design of new CAR-T agents targeting the tumor microenvironment (e.g., CAR-T cells targeting tumor stroma, extracellular matrix, suppressive cells, and cytokines/chemokines) and CAR-T boosting vaccines [4,5]. Since these strategies have demonstrated that it may be possible to overcome the mentioned biological barriers and increase the efficacy of CAR-T therapy for the treatment of solid tumors, the anti-JAG1 scFvs we developed may contribute to the success in solid tumors using such approaches.
To address the concern raised by the reviewer, this information was included in the present version of the manuscript (lanes 594 – 616).
Comment 2: The work is impressive in the sense of strategy. The result is expected, meaning is not really a novel avenue to solve the problem. I also imagine the authors could have developed ScFv proteins for any other type of target.
Reply to comment 2: We thank the reviewer for his appreciation of the strategies we employed in this study to characterize the anti-JAG1 scFvs in the antibody and CAR-T cell modalities.
We understand the reviewer’s point of view that we could have also developed scFv against other cancer relevant proteins. However, we have focused on JAG1 due to its well-established oncogenic role in various aggressive solid tumors as well as our previous expertise and track record in oncology research using ligands of the Notch1 pathway. The scFv identified here can, for instance, provide to the community additional tools to overcome one of the main limitations in addressing solid tumors, which as the reviewer highlighted, is the heterogeneous distribution of potential targets in the tumors. Together with other relevant cancer targets that have been extensively explored in the context of immunotherapy (e.g. HER-2, EGFR, mesothelin) [1], these scFv can be applied to combinatorial antigen-recognition circuits (e.g. SynNotch system) or dual-target CAR approaches to improve both the efficacy and safety, by minimizing on-target off-tumor effects [6,7]. Thus, despite not presenting a novel strategy to solve the problem of the limited efficacy of CAR-T targeting in solid tumors, as discussed in the previous comment, the results we obtained in this study clearly show that our anti-JAG1 scFvs might be good candidates for the development of cell therapies targeting JAG1-positive tumors. This information has been included to the manuscript's revised version in the passage referring to combinatorial methods detailed in the response to comment 1.
Comment 3: I found the introduction lacks citations. The sentences between lines 42 to 49. Similarly, between 65 to 70.
Reply to comment 3: As suggested by the reviewer, citations have been added in the indicated places in the updated version of the manuscript.
Comment 4: The authors refer much to citation #26, published by them. Righteously so. I would like to see a few more references to previous articles through the methods as well. But I understand that is hard to do sometimes.
Reply to comment 4: In agreement with the reviewer’s suggestion, we have now introduced a few more references to previous articles in the methods section.
Comment 5: In the conclusions authors write:
- Conclusions 558
Using a recombinant truncated human JAG1 protein, containing the JAG1 regions 559 involved in Notch binding/activation, we selected anti-JAG1 scFv binders from phage dis- 560 play libraries. This sentence seems to come from things the authors did in reference#26. If this is correct they did not do this work on this manuscript.
Reply to comment 5: The reviewer is correct that the recombinant human JAG1 protein used to select specific anti-JAG1 scFv by phage display was developed in scope of our previous publication. As such, this information has now been removed from the conclusion section in the present version of the manuscript.
References:
- Patel, U.; Abernathy, J.; Savani, B.N.; Oluwole, O.; Sengsayadeth, S.; Dholaria, B. CAR T cell therapy in solid tumors: A review of current clinical trials. EJHaem 2022, 3, 24-31, doi:10.1002/jha2.356.
- Siddiqui, R.S.; Sardar, M. A Systematic Review of the Role of Chimeric Antigen Receptor T (CAR-T) Cell Therapy in the Treatment of Solid Tumors. Cureus 2021, 13, e14494, doi:10.7759/cureus.14494.
- Hull, C.M.; Maher, J. Approaches for refining and furthering the development of CAR-based T cell therapies for solid malignancies. Expert Opin Drug Discov 2021, 16, 1105-1117, doi:10.1080/17460441.2021.1929920.
- Safarzadeh Kozani, P.; Safarzadeh Kozani, P.; Ahmadi Najafabadi, M.; Yousefi, F.; Mirarefin, S.M.J.; Rahbarizadeh, F. Recent Advances in Solid Tumor CAR-T Cell Therapy: Driving Tumor Cells From Hero to Zero? Frontiers in immunology 2022, 13, 795164, doi:10.3389/fimmu.2022.795164.
- Liu, G.; Rui, W.; Zhao, X.; Lin, X. Enhancing CAR-T cell efficacy in solid tumors by targeting the tumor microenvironment. Cell Mol Immunol 2021, 18, 1085-1095, doi:10.1038/s41423-021-00655-2.
- Roybal, K.T.; Rupp, L.J.; Morsut, L.; Walker, W.J.; McNally, K.A.; Park, J.S.; Lim, W.A. Precision Tumor Recognition by T Cells With Combinatorial Antigen-Sensing Circuits. Cell 2016, 164, 770-779, doi:10.1016/j.cell.2016.01.011.
- Hyrenius-Wittsten, A.; Su, Y.; Park, M.; Garcia, J.M.; Alavi, J.; Perry, N.; Montgomery, G.; Liu, B.; Roybal, K.T. SynNotch CAR circuits enhance solid tumor recognition and promote persistent antitumor activity in mouse models. Sci Transl Med 2021, 13, doi:10.1126/scitranslmed.abd8836.
Reviewer 2 Report
Chimeric antigen receptor (CAR) T-cell therapy originally developed for the treatment of hematologic malignancies has the huge scientific, clinical and public interest. Though there are limitations for use of (CAR) T-cell technology in solid tumors treatment due to the lack of suitable target antigens, poor availability of the target antigen for CAR-T, and heterogeneous pattern of tumor antigen expression. CAR-T cell therapy is rapidly developing field and has tremendous potential for many aggressive tumors. The authors of this manuscript report identification of new anti-JAG1 scFv representing a promising candidate for the development of cell therapies against JAG1-positive tumors. This manuscript is clearly written. The aim and scope of the study explained well. Introduction is quite comprehensive and highlighted work importance as well as its significance towards future prospective. Research was performed at a high methodological level. Authors provided well explained interpretation of results and discussion.
Just minor spell check is required. (for example, 421 line: instead of CD218 linker should be 218 linker).
Author Response
Reply to the reviewer: We would like to thank the reviewer for the overall positive comments on our manuscript.
Comment: Just minor spell check is required. (for example, 421 line: instead of CD218 linker should be 218 linker
Reply to comment:
As suggested, the manuscript was revised to correct any mistakes as noted by the reviewer.
Reviewer 3 Report
The manuscript by Silva et al. addresses the suitability of Notch ligand JAG1 as a cell therapy target for ARG1-positive tumors. Anti-JAG1 scFvs were screened from phage display. The authors tested the affinity, and specificity of the scFvs and constructed anti-JAG1 CARs. The CARs showed great recognition and killing of JAG1 expression cells. The paper is sound and very clearly written, but it could be great if the authors can show the anti-JAG1 CAR-specific killing in mouse tumor models and the on-target off-tumor activity which the reader will definitely want to see.
-In Figure 1b, the legend shapes are not consistent with the shapes in the figure.
-For Figure 2a, can you explain why different linkers and sequences of the VH-VL/VL-VH were used? Either one of the two could influent the efficacy of the CAR T cell. When you construct the groups of CARs, you change those two variables at the same time.
Author Response
Comment 1: The paper is sound and very clearly written, but it could be great if the authors can show the anti-JAG1 CAR-specific killing in mouse tumor models and the on-target off-tumor activity which the reader will definitely want to see.
Reply to comment 1: We totally agree with reviewer that readers with interest in this research area would want to see in vivo data. However, given the substantial time and resources needed to address the on-target off-tumor activity of the anti-JAG1 B5(VH-VL)-3G CAR in relevant mouse tumor models, we plan to perform these experiments in follow up studies. Namely, in in vivo studies using this anti-JAG1 B5(VH-VL)-3G CAR to show this effect, and most importantly, to show the exclusive killing of JAG1-expressing cancer cells by this CAR. For the precise killing of tumor cells, we plan to use engineered T cells with Synthetic Notch receptors to allow CAR activation exclusively upon recognition of JAG1 expressed in cancer cells. Additionally, due to the biologic barriers of solid tumors, that limit the success of CAR-T cell therapy [1] , we plan to engineer the CAR-T cells with tools that would permit them to penetrate the tumour microenvironment and hamper its immunosuppressive action, aiming to increase the efficacy of the anti-JAG1 CAR in the chosen solid tumor models.
Nevertheless, due to the well-established oncogenic role of JAG1 in various aggressive solid tumors (e.g., breast, colorectal, gastric, prostate, hepatocellular) we believe the work we present here contains significant novelty and is relevant for the cell therapy research communities, since it may increase the toolbox of scFvs and relevant targets with potential for the development of therapies against aggressive solid tumors.
Comment 2: In Figure 1b, the legend shapes are not consistent with the shapes in the figure.
Reply to comment 2: This has been corrected in the Figure 1b.
Comment 3: For Figure 2a, can you explain why different linkers and sequences of the VH-VL/VL-VH were used? Either one of the two could influent the efficacy of the CAR T cell. When you construct the groups of CARs, you change those two variables at the same time.
Reply to comment 3: As pointed out by the reviewer, and mentioned in the manuscript, the functionality of CAR-T cells is influenced by the different CAR components and their order in the molecule, amongst other factors. This is the reason why we chose to generate several anti-JAG1 CARs with the VH and VL regions in different positions as well as evaluate different linkers. For the scFv of both J1.B5 and J.F1 antibodies we decided to generate CARs using VH-VL order and a (GGGGS)3 linker because this was the scFv format present in the clones selected by phage display that effectively bound the recombinant JAG1 proteins as demonstrated by phage-ELISA (Figure S1b). By doing so, we reasoned that these scFvs would also recognize the respective target antigen in the context of a CAR protein. For the scFv of J1.F1 (our lead Ab) we also decided to generate CARs with the scFv and the 218 linker in the configuration of therapeutic anti-CD19 CARs (i.e., VL-218-VH) to test whether this would lead to a superior CAR-T cell effector function.
As stated in the manuscript, the different data obtained with the generated J1.F1 CARs, in the activation of Jurkat cells in the presence of target HepG2 cells, are likely due to the different VH/VL positioning and linkers in the respective scFvs, which dictate the scFv conformational structure and ultimately the ability to bind the target antigen.
These two variables were separately introduced into the CAR constructs containing all the other CAR components/blocks during the cloning procedure.
The manuscript has now been edited to ensure this information is more clear.
Reference
- Safarzadeh Kozani, P.; Safarzadeh Kozani, P.; Ahmadi Najafabadi, M.; Yousefi, F.; Mirarefin, S.M.J.; Rahbarizadeh, F. Recent Advances in Solid Tumor CAR-T Cell Therapy: Driving Tumor Cells From Hero to Zero? Frontiers in immunology 2022, 13, 795164, doi:10.3389/fimmu.2022.795164.
Reviewer 4 Report
Authors have presented an interesting manuscript entitled “Novel scFv against the Notch ligand JAG1 suitable for the development of cell therapies towards JAG1-positive tumors”. The article is original, well structured; easy to read with main emphasis on the exploration of anticancer therapy via targeting notch signaling pathway with novel antibodies construct. Authors have described the concept to a greater extent but the manuscript still need some minor corrections before publishing.
· Authors are suggested to explain the concept of study design in a better way in introduction section.
· Explain the reason of targeting only Jag1, why not other potential target of Notch signaling.
· Improve the dimension/resolution of figures so that these can be clearly visible.
· Authors must present the peak form result of GC-MS analysis. Authors can include this result as supplementary file.
· Authors must provide the reason of selection of biomarkers for gene and protein expression analysis. How these biomarkers could justify the study design, for example bcl2 and caspase-3 expression could not justify apoptotic activity of extract alone.
· Is there any cytotoxicity reported of this antibody? Authors must explain the cytotoxicity part in result and discussion section.
· Conclusion part should be improved by addition of future directions.
· Abbreviations should be clearly stated throughout the manuscript
Author Response
Comment 1: Authors are suggested to explain the concept of study design in a better way in introduction section.
Reply to comment 1: As per this reviewer’s suggestion the introduction section of the manuscript has been edited to clearly clarify the concept of study design.
Comment 2: Explain the reason of targeting only Jag1, why not other potential target of Notch signaling.
Reply to comment 2: In previous work we developed antibodies against the Notch ligand DLL1 and identified an anti-DLL1 antibody (Dl1.72) with anti-tumor efficacy against estrogen receptor-positive breast cancer [1]. In the current manuscript we focused on targeting JAG1 because it is a well-established oncogene in various aggressive tumors associated with poor overall survival. Given the multi-functional roles ascribed to JAG1 in cancer biology (promotion of cancer cell survival, proliferation, epithelial-to-mesenchymal transition, metastasis, stem cancer cells expansion, tumor-associated angiogenesis, inhibition of tumor-specific immunity, drug resistance, and tumor recurrence) we believe it should be considered a major target for cancer therapy and thus be explored in the context of immunotherapy. With this work our aim was to identify novel specific anti-JAG1 scFvs with potential for the development of therapeutic products for the treatment of aggressive solid tumors.
Comment 3: Improve the dimension/resolution of figures so that these can be clearly visible.
Reply to comment 3: To address the reviewer´s concern, previous Figure 1 has now been split up into 2 new Figures in the present version of the manuscript, to allow better visualization.
Comment 4: Authors must present the peak form result of GC-MS analysis. Authors can include this result as supplementary file.
Reply to comment 4: As per this reviewer’s suggestion we have now included in the revised version of the manuscript a new supplementary figure, now labelled as Figure S2, showing the peak form result from the size exclusion-high-performance liquid chromatography (SE-HPLC) analysis of each of the antibodies. For clarification, no GC-MS analysis was conducted in this study.
Comment 5: Authors must provide the reason of selection of biomarkers for gene and protein expression analysis. How these biomarkers could justify the study design, for example bcl2 and caspase-3 expression could not justify apoptotic activity of extract alone.
Reply to comment 5: For the reviewer’s information, we have not performed any gene expression analysis for biomarker selection. However, we performed protein expression analysis to evaluate the expression of our CAR constructs in Jurkat cells and JAG1 expression in the various cell lines used in this study (new Figure 3b, Figure S1d, and new Figure S3a).
Comment 6: Is there any cytotoxicity reported of this antibody? Authors must explain the cytotoxicity part in result and discussion section.
Reply to comment 6: In this work, we thoroughly characterized the antibodies we produced using different methodologies and no associated toxicity was detected in any of the assays we performed. In the case of the anti-JAG1 CARs, significant killing activity was observed only in the presence of JAG1-expressing cancer cells (new Figure 5g). This information in now included in the revised version of the manuscript.
Comment 7: Conclusion part should be improved by addition of future directions.
Reply to comment 7: As requested, this information was added into the conclusion´s section of the manuscript.
Comment 8: Abbreviations should be clearly stated throughout the manuscript
Reply to comment 8: We have addressed this concern in the revised version of the manuscript.
Reference
- Silva, G.; Sales-Dias, J.; Casal, D.; Alves, S.; Domenici, G.; Barreto, C.; Matos, C.; Lemos, A.R.; Matias, A.T.; Kucheryava, K.; et al. Development of Dl1.72, a Novel Anti-DLL1 Antibody with Anti-Tumor Efficacy against Estrogen Receptor-Positive Breast Cancer. Cancers 2021, 13, doi:10.3390/cancers13164074.